# The 360° Performance System in Team Sports: Is It Time to Design a “Personalized Jacket” for Team Sports Players?

**DOI:** 10.3390/sports9030040

**Published:** 2021-03-17

**Authors:** Igor Jukic, Julio Calleja-González, Francesco Cuzzolin, Jaime Sampaio, Francesc Cos, Luka Milanovic, Ivan Krakan, Sergej Ostojic, Jesús Olmo, Bernardo Requena, Nenad Njaradi, Roberto Sassi, Mar Rovira, Baris Kocaoglu

**Affiliations:** 1Faculty of Kinesiology, University of Zagreb, 10110 Zagreb, Croatia; igor.jukic@kif.unizg.hr (I.J.); luka.milanovic@kif.unizg.hr (L.M.); ivankrakan@gmail.com (I.K.); 2Biotrening Ltd., 10000 Zagreb, Croatia; 3Faculty of Education and Sport, University of Basque Country (UPV/EHU), 01007 Vitoria-Gasteiz, Spain; 4Technogym SpA, 47521 Cesena, Italy; fcuzzolin@technogym.com; 5Research Centre in Sports Sciences, Health Sciences and Human Development, CIDESD, 5001-801 Vila Real, Portugal; jaimesampaio@icloud.com; 6National Institute of Physical Education (INEFC), University of Barcelona, 08038 Barcelona, Spain; cosfrancesc@gmail.com; 7Manchester City Football Club, Manchester M11 4TS, UK; 8Faculty of Physical Education and Sport, University of Novi Sad, 21000 Novi Sad, Serbia; sergej.ostojic@chess.ed.rs; 9Center for Health, Exercise and Sport Sciences, 11000 Belgrade, Serbia; 10Football Science Institute, 18016 Granada, Spain; jesusolmo@me.com (J.O.); bernardorequena@footballscienceinstitute.com (B.R.); 11Football Club Deportivo Alavés, 01007 Vitoria-Gasteiz, Spain; nenad.njaradi@gmail.com; 12Football Club Juventus, 10151 Torino, Italy; r.sassi@gmail.com; 13Tecnocampus, Universitat Pompeu Fabra, Grup de recerca en Activitat Física, Rendiment i Salut (AFIRS), Av. d’Ernest Lluch, 32, 08302 Mataró, Spain; marrovira@hotmail.com; 14Faculty of Medicine, Acibadem University, Küçükbakkalköy, Kayışdağı Cd., 34755 Ataşehir/İstanbul, Turkey; bariskocaoglu@gmail.com

**Keywords:** team sports, performance, players

## Abstract

Elite performance in team sports attracts the attention of the general public. In particular, the best players became incredibly skilled and physically powerful, which is a fact that potentiates the delivery of a product that is considered attractive, exciting, and competitive. Not surprisingly, this is a very valuable product from an economic and social standpoint; thus, all sports professionals are extremely interested in developing new procedures to improve their sports performance. Furthermore, the great interests of the various stakeholders (owners, chief executive officers (CEOs), agents, fans, media, coaches, players, families, and friends) are one of the main reasons for this development under the sports science umbrella and the accompanying sports industry. All their personal performances should be coordinated and put into practice by the sports team. In this scientific and applied study, we primarily dealt with the individual treatment of players in order to improve their personal performance and, consequently, the team’s sporting performance.

## 1. Introduction

Elite performance in team sports attracts attention from the general public [1,2]. In particular, the best players became incredibly skilled and physically powerful that potentiates the delivery of a product that is considered attractive, exciting, and competitive. Not surprisingly, this is a very valuable product from an economic and social standpoint; thus, all sports professionals are extremely interested in developing new procedures to improve their sports performance. Furthermore, the great interests of the various stakeholders (owners, chief executive officers (CEOs), agents, fans, media, coaches, players, families, and friends) provide one of the main reasons for this development under the sports science umbrella and the accompanying sports industry. In this sense, sport sciences can greatly help to improve athletic performance [3]. This is particularly noticeable in a time of major changes in the regular training and competition regime, such as that which occurred during the coronavirus disease 2019 (COVID-19) pandemic [4].

An elite player in team sports has a large number of characteristics that are in synergy with each other. The optimal levels of development of individual characteristics, as well as their mutual harmonization, are achieved by an integrated and personalized system of sports preparation [5]. The basic task of sports preparation is to enable players and teams to achieve the best sports performance in their competition. Elite sports achievements are reached through the long-term, systematic, and hard work of a large number of people who service athletes [6]. Sports coaches, strength and conditioning coaches, psychologists, medical doctors, physiotherapists, nutritionists, and sports analysts form multidisciplinary teams that base their work on interdisciplinary synchronization [7]. All these experts give their coordinated contribution to better optimize the sports teams’ system, as well as to the quality performance of each individual [8]. The contributions of each player to the team performance occur based on the wise leadership, management, and regulation of the head coach. The head coach, together with the general manager or sports director, selects the players, assigns them roles and responsibilities, determines the rules of conduct and action, and coordinates the professional staff members. At the same time, at the level of the sports organization management (club or federation), an interdisciplinary team of experts (ticket sales and sales manager, lawyer, accountant, community manager, marketing manager, graphic designer, administrators, insurance officer, etc.), led by the general manager and sports director, harmoniously provide optimal conditions for sports preparation for teams, players, and staff members [9]. In this way, at the same time, different but interconnected processes occur with the unique goal of achieving the best possible sports outcome (Figure 1).

One of the most important tasks of professional staff is to bring each player into the zone of their personal best performance. After that, all their personal performances should be coordinated and put into practice by the sports team. In this scientific and applied study, we primarily dealt with the individual treatment of players in order to improve their personal performance and, consequently, team sport performance.

## 2. Individual Characteristics of the Player

The individual characteristics of the players need to be identified multidimensionally, but in an integrated way [4]. In this sense, sports performance arises as a result of the integrative procedures of the competition, training, and recovery of an athlete through their sports development cycle (career) [10,11]. In order to embark on a synergistic process of integrating all the athlete characteristics, it is important to understand which characteristics describe an individual athlete profile (Figure 2).

### 2.1. Health

The fundamental value of human life is health. In elite sports, health is mainly observed through the availability of players for training and competition [2]. An absence of players due to injuries or disease seriously affects competitive results [12]. Availability is ensured primarily by syndrome prevention [13,14] and then by the rapid and safe return of players to regular training and competition [15,16]. At the same time, locomotor health has a high priority, namely, the most common cause of player absences are injuries in muscles, bones, tendons, fascias, and joints [17,18]. Furthermore, it is very important to take care of the player’s immunity and respiratory and metabolic health. High levels of competitive stress, depression, demanding continuous travel fatigue, and significant exhaustion compromise a player’s immune system, which can lead to a variety of diseases [19]. In this way, mental health must be considered [20].

Due to all of the above, it is necessary to create a health profile of each player, which contains:−A history of injuries and illnesses;−Locomotor deficits;−Immune and metabolic deficits;−Personalized protocols/guidelines to minimize injury risk;−Medical interventions;−Surgical and conservative treatments;−Rehabilitation processes.

Every team sport has its typical health threats; therefore, it is important to take into account information about the frequency, risks, and mechanisms of injury and disease in a particular sport. Athletes are supervised by a medical team [20], which consists of team physician staff with various specialties (orthopaedists/traumatologists, sports medicine specialists, dentists, ophthalmologists, internists, etc.) and physiotherapists, osteopaths, kinesitherapists, laboratory technicians, and nurses, among others. In this regard, the club usually employs multiple doctors who have their own medical team leader, sometimes under a referent hospital in the same city.

### 2.2. Age and Gender

Age and gender are given and (mostly) unchanging characteristics of athletes. Nevertheless, respect for age characteristics and gender specificities can to a large extent make the process of sports preparation safer and more efficient [6,21,22]. For example, common injuries in females (e.g., female triad). When registering and monitoring the age of athletes, it is important to establish the following:−Chronological age;−Biological age (especially in young athletes);−Metabolic age;−Sports age (years spent in organized sport).

It is also important to link health, training, and competition events in an athlete’s previous career with the current situation. The sporting longevity of elite athletes is strongly related to current habits, behaviours, and the social environment, but also to previous experiences and events throughout a sports career. We are witnessing that sports careers are lasting longer (up to 35–40 years) [23], probably due to the improved system of personalized preparation and the behaviours of athletes based on epigenetic pathways [24].

Information on the age and gender of athletes is respected and used by all members of the professional team to organize personalized training and medical, nutritional, and psychological care.

### 2.3. Fitness

An adequate fitness profile is a key contribution to the optimal expression of sports skills [4]. In fact, players with solid athleticism have a much wider spectrum of possibilities to develop and fine-tune their technical and tactical skills available to them [1]. Furthermore, well-trained athletes are less prone to injuries and recover much quicker after intense training and competitive loads [25], thereby likely being available to play a lot of matches throughout the season. In the fitness profiling of players, it is important to recognize the following:−Mobility and stability of the locomotor system;−Synergy/balance of agonist/antagonist muscles and muscle chains;−Energy systems;−Neuromuscular abilities;−Sports discipline;−Age;−Gender.

Based on the profile and identified deficits, preventive–corrective, energy, and neuromuscular programs can be created to overcome the identified needs. All capacities need to be optimized, but not necessarily maximized, as fitness qualities are a function of sporting skills. Therefore, it is important to align fitness programs with the player’s health status, age, gender, training history and culture, sports characteristics and position in the game, and the current level of training.

The fitness training of players is planned, programmed, and carried out by fitness strength and conditioning specialists, but also in part by sports coaches, kinesitherapists, osteopaths, and physiotherapists. Different tools should be used to quantify these factors using standard tests [26].

### 2.4. Body Shape

The shape and structure of the player’s body should be in line with the needs and requirements of the sport and with the individual athlete’s needs [26]. This applies in particular to the following:−The size of the body and its parts;−Relative proportions between body parts;−Body composition:○Muscles, subcutaneous fat, and bones;○Internal muscle structure (the types of muscle fibres and muscle architecture);−Individual somatotype.

The shape and structure of a player’s body can be changed by training, diet, and external stimuli [27,28,29]. Depending on the goal, training and nutrition programs are created that are aimed at optimizing the shape and structure of the body in accordance with the requirements of the sport and the individual characteristics of the players. Strength and conditioning coaches and nutritionists take care of the shape and structure of the body.

### 2.5. Trainability and Learnability

The same training program can result in different responses from different players [30,31]. The ability of an athlete to learn different movement structures and to apply them in training and competitive situations is called learnability. The player’s technical and tactical performance depends on this ability. On the other hand, the enhancement of a player’s abilities (both energetic and neuromuscular) based on applied training programs is called trainability [32]. Different players need a different combination of content and load for the same or targeted performance [33]. The player’s physical development and form depend on this ability. These two abilities are essential for the work of primarily sports coaches and strength and conditioning coaches, based on standard tests [1].

### 2.6. Sports History and Culture

In order to be able to draw conclusions about an athlete’s current condition that are reached using diagnostic procedures, it is important to know the following:−What kind of sports environment and culture the player comes from?−What training process has the player gone through during their career so far?−What kind of training processes has the player had in the last few months?−What is the competitive history of the player?−Previous personal experiences from different enviroments and cultures.

The information obtained from the sports profiles of athletes serves as a prerequisite for designing individual and team sports preparation programs, which are prepared and implemented by sports and strength and conditioning coaches [4].

### 2.7. Recovery

Player tolerance to different types of fatigue during and after exercise and the ability to recover within and after exercise are the bases for creating a recovery profile [34,35]. The recovery profile includes the following:−Individual tolerance to different types of fatigue;−Dynamics of recovery during training and competition;−Dynamics of recovery after training and competition;−The most appropriate means and methods of recovery during and after training and competition;−Optimal doses of selected agents and methods of recovery.

Each player should have their own recovery profile that serves as the basis for creating personalized recovery protocols [8,36]. The development of recovery profiles and the implementation of the recovery process are carried out by physicians, sports scientists, strength and conditioning coaches, nutritionists, psychologists, and physiotherapists [8,37].

### 2.8. Mindset

The mental characteristics of a player determine their behaviour in life, training, and competition. If a player’s way of thinking and behaving is in line with the requirements of elite sport, the likelihood of their success increases [38]. Since elite sports often place extreme demands on the player, the player’s mindset must be adapted to such conditions. The following characteristics are especially important in elite team sports:−Appropriate motivation;−Emotion control;−Cognitive mobility;−High focus;−Communication skills;−Self-discipline.

Sports psychology has effective tools and methods in its portfolio to improve all of these characteristics [39]. In addition to psychologists, sports coaches are involved in the work on improving the player’s psychological characteristics. Psychological and psychosocial interventions have a moderately positive effect on sports performance [40].

### 2.9. Lifestyle

Since the player spends most of the daytime in their own environment and organization, the control and interventions in the player’s lifestyle occupy an increasingly important place [41]. Lifestyle segments that are especially important for the integral readiness of athletes are as follows:−Duration and quality of sleep;−Adequate nutrition;−Quality hygiene habits;−Family life;−Social life;−Hobbies;−Rest;−Housework and procurement;−Fun and entertainment;−Intimate life;−Consumption of harmful substances;−Self-health.

Coaches, a psychologist, a nutritionist, and a doctor, as well as close family members or friends, can take part in controlling a player’s lifestyle.

### 2.10. Skills

The player’s competitive success depends upon their technical and tactical skills [6,42,43]; therefore, there is no surprise that the largest proportion of training work is allocated to technical and tactical training [6]. The player’s decisions in competitive conditions are the result of a whole conglomeration of influences that take place within the system of sports preparation. Therefore, it is important to take into account the personalization of different aspects of sports preparation, including:−The position in the game/team;−Retrospective and prospective analysis of competitive performance;−The process of learning, both individually and collectively;−Expertise in tactical training.

In the last few years, there has been a substantial increase in popularity from the skill acquisition perspectives that have been sustained by the ecological dynamics approach [44] and there are already several examples of successful practical applications [45,46]. This needed update might be the precious help that is required to reach the individual needs of each player [4].

Therefore, it is important to dedicate a certain amount of total training time to improving individual sports techniques and skills and to provide individual analysis of competitive performance. Sports coaches are responsible for planning, programming, and controlling technical–tactical preparation; however, skill acquisition specialists are more frequently included among the coaching staff in order to provide new expertise on developmental pathways for players and the optimization of short-, mid-, and long-term learning in training sessions.

## 3. Creating the 360° Personal Jacket Performance System

The complexity of a personalized system of sports preparation in team sports lies in the synchronization of work and synergistic actions of all experts in the sports organization, which gathers a larger number of players [47,48]. Furthermore, each player has a number of characteristics to identify, analyse, and monitor. Some characteristics are in deficit; therefore, they should be brought to an acceptable and optimal state. However, other characteristics are at an acceptable or above-average level and should be further improved and their comparative value emphasized. In addition, all individuals (players and experts) need to be teamed up in order for the final result of the sports team to be successful. Therefore, it is important to implement a clear structure and hierarchized network within the expert team and to define roles, rules, and responsibilities for each member of the expert team/staff (Figure 3).

It is especially important to determine the method of communication within the expert team, i.e., the communication algorithms and procedures [49]. This is important because of the need to rationalize the data collections, processing, and communication of large amounts of information that describe the daily state of readiness and activities of each athlete. Therefore, there is a clear need to have optimized solutions that are designed by data architects, data engineers, and data analysts in a way that centralizes all data to be processed such that it is later available according to the personal needs of each expert staff member or even the players.

Creating a 360° Personal Jacket Performance System has predefined phases, which can be presented in four major steps:−Personalized history;−Personalized diagnostics;−Personalized goals;−Personalized 360° programs;−Personalized monitoring.

All phases of the system are implemented in each of the system performance sectors. The general performance system (strength and conditioning, sports medicine, nutrition, psychology, recovery, lifestyle interventions, performance analytics) has the role of supporting the system of specific preparation (technical–tactical preparation).

One possible approach to collecting, processing, and using player data is the matrix approach. All personalization steps in the system are carried out in each of the sectors of operation and are presented together in a matrix (Table 1). The matrix for each player contains the basic characteristics of the athlete (Figure 2), basic areas of work (Figure 4), and basic operational procedures (history, assessment, goals, programs, monitoring). The matrix is available to all team members and the final appearance of the matrix and all interventions according to the individual player needs are finally approved by the head coach with the suggestion of the head of sports preparation (head of performance). Communication containing proposals and approvals takes place in the main (all sector leaders present), coordination (the head coach with the head of performance or the head of performance with the sector leaders), and/or sectoral meetings (the members of one sector present).

Each segment of the model has additional detailed elaboration, which is created by an expert for a particular area. All information about each player is stored in a web cloud database. At the beginning of each season and the end of each cycle of preparation or competition, the expert team, led by the head of performance, presents the current status, goals, and interventions to the head coach. Furthermore, in the daily meetings, the performance team discusses all topics and issues related to acute interventions. The head of performance then presents the most important operational details to the head coach. At the same time, members of the performance team are in contact with the players with whom they carry out interventions on a daily basis.

One of the possible models of daily work organization is presented in Table 2 and Table 3.

Training and recovery interventions are personalized in the following forms of daily work:Pre-formance—programs of individual preparation for training that precedes the team warm-up. This preparation is based on the individual needs of the athletes, but also the requirements of the upcoming training/match. Such programs can last from 10–60 min and can be conducted by sports coaches, analysts, strength and conditioning coaches, medical doctors, nutritionists, physiotherapists, and sports psychologists.In-formance—individual programs that are implemented during training sessions. These programs can be conducted during special training periods that are provided for individual stimuli or with players not involved in team programs. The total duration of the in-formance program occupies 5–20% of the total duration of the training and is mainly conducted by sports and strength and conditioning coaches, skill acquisition experts, and physiotherapists.Post-formance—individual programs that are conducted after training, between the end of team training and the recovery and regeneration protocol. The total duration of these programs is between 10 and 45 min and it is conducted by sports coaches, analysts, strength and conditioning coaches, medical doctors, nutritionists, physiotherapists, and sports psychologists.Extra-formance—programs that are conducted outside of regular team training, usually in the second part of the day in relation to team training. These programs can take place under the guidance of club experts (sports coaches, analysts, skill acquisition, strength and conditioning coaches, medical doctors, nutritionists, physiotherapists, and sports psychologists) or in the private arrangement of the players, with their personal experts. The total duration of these programs is between 30 and 90 min.

A personalized approach to prepare athletes for elite performance is a complex and demanding job. Such work takes place daily and continuously throughout the competition season. The daily organization of work in which special attention is paid to the individual needs of athletes requires quality synchronization of all members of the professional team. Such a daily program is planned in advance and is based on both the acute (based on daily parameters) and chronic (based on the parameters of periodic testing) statuses of the players. In doing so, the chronic statuses and goals are the basis for the development of daily training programs and the acute statuses determine the need for the current corrections of training programs. For each of the parts of the daily program (pre-formance, team training, in-formance, post-formance, extra-formance), the protocols of individual interventions are made by different members of the professional team/staff. An example of personalization within the complete daily program is seen in Table 4.

## 4. Conclusions

The 360° Personal Jacket Performance System aims to enable the maximal use of all personal potentials of players in team sports in order to improve the performance of the team. The implementation of this system includes a multidisciplinary team of experts that is synchronized and led by a leader (director of performance) with the purpose of optimizing the processes of the team and the individual sports preparation of players. The synergistic action of all experts enables the team and each individual player to improve their sports performance.

With the whole structure and organization of a personalized approach to improving player performance, the main protagonist of this system of work (the 360° Personal Jacket Performance System) is the player. The player’s understanding of the need for such a system, strong motivation, self-discipline, and commitment are the key prerequisites for the success of this system. Furthermore, every player should consistently think about this system. This way of thinking and behaving gives the player the opportunity to stay healthy, improve their competitive performance, and prolong their career.

## 5. Practical Applications

This scientific document provides an important first approach toward the progress of performance knowledge for team sports athletes. This first approach can be useful for practitioners in order to improve the performance in these sports.

## 6. Future Lines

Each particular sport needs more research in order to better understand holistic performance.

## Figures and Tables

**Figure 1 sports-09-00040-f001:**
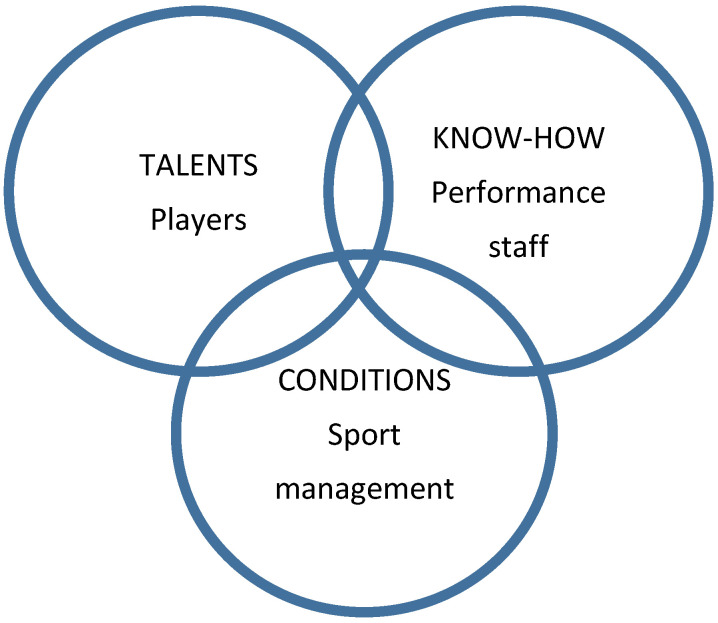
Structure and inter-relationships of the main stakeholders in sporting organizations.

**Figure 2 sports-09-00040-f002:**
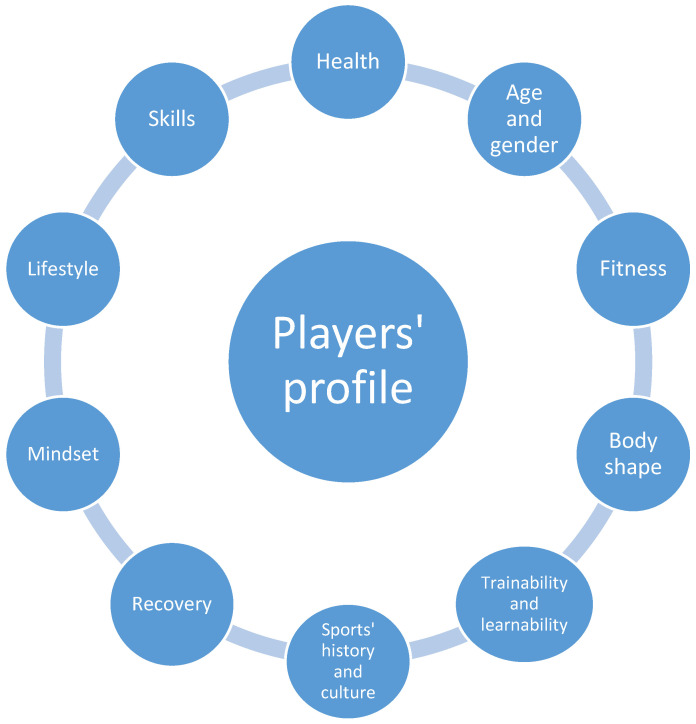
Individual profile of player in a team sport.

**Figure 3 sports-09-00040-f003:**
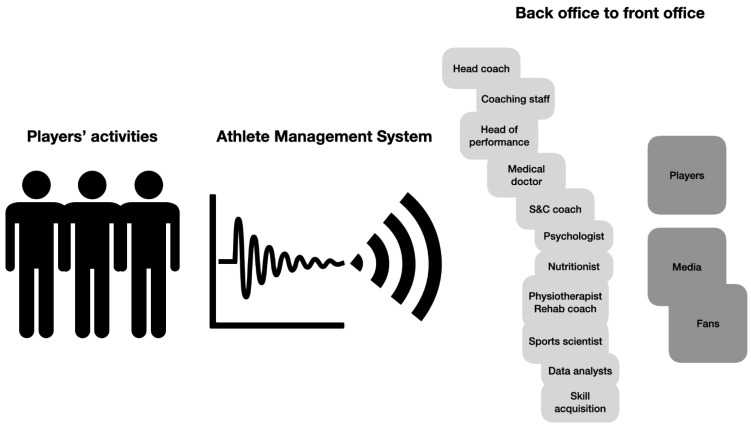
Structure and hierarchy of the coaching and performance staff in a sporting organization. S&C: strength and conditioning.

**Figure 4 sports-09-00040-f004:**
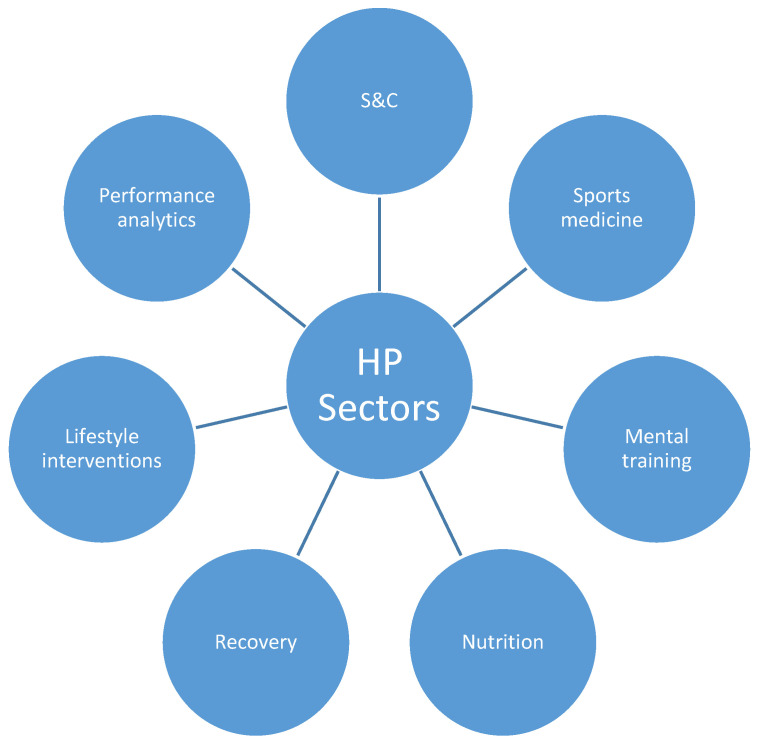
High-performance (HP) sectors in a sports organization.

**Table 1 sports-09-00040-t001:** An example 360° Personal Jacket matrix.

	Performance Analysis	Health	Strength and Conditioning	Mental Training	Nutrition	Recovery	Lifestyle
**History**	12 years in sport, 4 years at the elite level	ACL—L 6 years ago	WL—4 RT—8 SAQ—6	3	5	5	4
**Assessment**	D—6 O—8 T—7	RLM—4 RLJ—7 RLI—3	MS—7 EN—6 NMS&P—7 NMSAQ—5	M—9 EC—4 F—5 CM—6	DNI—5 SS—62 Tanita BF—15%	HRR—7 HRV—6	S—6 NH—5
**Goals**	D—7 O—8 T—8	RLM—2 RLJ—6 RLI—2	MS—8 EN—8 NMS&P—8 NMSAQ—7	M—9 EC—6 F—7 CM—7	DNI—6 SS—58 Tanita BF—13%	HRR—8 HRV—7	S—7 NH—7
**Programs**	TeTa team and personal training	PRECOR	PRECOR PENT PNMS&PT PNMSAQ T	PMT	PNP	PRP	SII NHII
**Monitoring**	GPS TT GPS TR	IIOF	MS—FMS EN—GPS NMS&P—CMJ, TDL NMSAQ, 20 mS, TT	PTO	SS Tanita BF	HRR HRV	WQ

Legend: Grades 1 (bad)–10 (excellent); D—defence; O—offence; T—transition; RLM—risk level (muscles); RLJ—risk level (joints); RLI—risk level (immunity); MM—movement mechanics; WL—weight lifting; RT—resistance training; SAQ—speed, agility, and quickness technique; MS—mobility/stability; EN—bioenergetic capacities; NMS&P—neuromuscular strength and power abilities; NMSAQ—neuromuscular speed, agility, and quickness abilities; M—motivation; EC—emotional control; F—focus; CM—cognitive mobility; GPSTT—technical–tactical GPS data; GPS TR—GPS tracking data; DNI—deep nutritional interview; SS—skinfold sum; Tanita BF—Tanita bioelectric impedance scale body fat assessment; HRR—heart rate recovery test; HRV—heart rate variability test; S—sleep; NH—nutrition habits; IIOF—injuries and illnesses occurrence form; MS–FMS—mobility/stability—functional movement screening; ENGPS—energetics GPS tracking data; CMJ—counter movement jump; TDL—trap dead lift; 20 mS—20 m sprint; TT—T-test; PTO—psychological training observation; PRECOR—preventive–corrective program; PENT—personalized energetic training; PNMS&PT—personalized neuromuscular strength and power training; PNMSAQT—personalised neuromuscular speed, agility, and quickness training; PMT—personalized mental training; PNP—personalized nutrition program; PRP—personalized recovery protocols; SII—sleep improvement intervention; NHII—nutrition habits improvement intervention.

**Table 2 sports-09-00040-t002:** Model of a daily schedule (for players) that includes personalized programs (* marked).

Time	Activity
−90 min	Arrival to training
−85 min	Wellness questionnaire and weighing *
−75 min	Breakfast *
−60 min	Team’s meeting with the coaching team
−50 min	Personalized assessment *
−45 min	Individual meetings with staff members and the physiotherapist *
−30 min	Pre-formance *
−5 min	Arrival to the pitch/court
0 min	Team’s training kicks off
0–90 min	In-formance (personalized part of team’s training) *
90 min	End of team’s training
+1 min	Post-formance *
+20 min	Recovery
+45 min	Rate of perceived exertion (RPE) *
+50 min	Shower
+75 min	Meal *
30–60 min	Extra-formance *

**Table 3 sports-09-00040-t003:** Model of daily schedule (for staff) that includes personalized programs.

Time	Activity
−120 min	Arrival to training
−110 min	Personal preparation for training
−100 min	Staff meeting
−90 min	Breakfast
−75 min	Preparation of workspace
−60 min	Team’s meeting with players
−50 min	Personalized assessment
−45 min	Individual meetings with players
−30 min	Pre-formance
−5 min	Arrival to the pitch/court
0 min	Team’s training kicks off
90 min	End of training
+1 min	Post-formance
+20 min	Recovery
+45 min	Cleaning the workspace
+60 min	Data management
+90 min	Staff meeting
+120 min	Shower
+135 min	Meal

**Table 4 sports-09-00040-t004:** Implementation of a training day—simulation of individual approach in team sports.

Player	Pre-Formance	Tactical Training	In-Formance	Post-Formance	Extra-Formance
Player 1	Hip mobility	100%	-	Bicycle capillarization	-
Player 2	Manual therapy	100%	-	High Intensity Interval Training (HIIT) short	Lower body strength
Player 3	Glute activation	80%	Dynamic hip stretching	Upper body strength	HIIT short
Player 4	Ankle mobility	100%	-	Core stability	Power—body mass weight
Player 5	Dynamic core stability	100%	-	HIIT short	-
Player 6	Glute strength	100%	-	Upper body strength	-
Player 7	Upper body strength	100%	-	Core stability	-
Player 8	Trap deadlift	100%	-	Bicycle capillarization	-
Player 9	Bicycle capillarization	60%	Bicycle capillarization	HIIT short	Lower body strength
Player 10	Ankle mobility	20%	Ankle rehab—manual therapy	Upper body strength	Physiotherapy
Player 11	Shoulder mobility	100%	-	Core stability	Lower body strength
Player 12	Electrostimulation	0%	Anterior Cruciate Ligament (ACL) rehab Knee mobility	Upper body strength	Physiotherapy
Player 13	Knee stability	100%	-	Hip mobility	-
Player 14	Lower back stability	0%	Lower back rehab—physiotherapy	Bicycle capillarization	Manual therapy

## Data Availability

No new data were created or analyzed in this study. Data sharing is not applicable to this article.

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
