# Peer review of "The 360° Performance System in Team Sports: Is It Time to Design a “Personalized Jacket” for Team Sports Players?"

_sports, 2021, doi:10.3390/sports9030040_

Round 1

Reviewer 1 Report

The manuscript deals with the he 360 Personal Jacket Performance System, to multidisciplinarily increase the performance of the players and in turn the team. 

General considerations regarding the manuscript:

Revise formatting, well-squared text, improve quality of all tables, improve quality of figure 3.

The references have different format and font size, adjust to the one requested in the journal. 

Author Response

Thanks so much for your effort. We have considered all your suggestions in order to improve the final text

1) We have rewieving fortmat and quality of figures. (Yellow colour in the text)

2) The reference have been revised as well (Yellow colour in the text)

In Advanced

King Regards

Reviewer 2 Report

The article is very interesting, but it provides very few practical implications and mostly describes very briefly the obvious aspects related to professional sport. I think it should be discussed with more details and insights. I understand that the article is already very long in the current version, but I have a problem finding information that provides new knowledge on this topic.

Line 34: capital letter

2.1. and 2.2.
In my opinion, the authors completely ignored the gender differences. e.g. female athlete triad syndrome. Moreover, I believe that aspects related to mental health (depression, perfectionism ..) and overtraining should be discussed. 

2.3 shouldn't fitness profiles also depend on the sports discipline, age, gender? What tool should be used to quantifying this?

2.4 do we have an influence on all these variables and can they are changed by training or diet? How do you evaluate all these variables?

2.5 how should you adjust the workouts? How to evaluate it?

2.6 It shouldn’t be a previous sports experience? And 2.7. This is the obvious information itself. What is the purpose of this paragraph? How to evaluate it? questionnaire? the velocity of movement? ​​or What?

2.8. and 2.9. Shouldn't some aspects related to the nutritional, health, etc. awareness be discussed?

Table 1.

"Mental training" "Nutrition" etc.? 5 is it good or bad? How is it evaluated? 

Table 2. trap deadlift as a preparation for training? Developing muscle strength after training? Shouldn't that be an extra performance?
Player 1 - power-resistance whole body?
Player 9 - isn't too much on lower-body?

Is it all possible to do in the real world? Is every athlete aware enough to do it himself or does he need specialized supervision? Should this plan be prepared each time for the next day, or maybe for a week?

Author Response

Thanks so much for your review and your effort. We have considered all your suggestions (good ideas, little details) in order to improve the final version of the text. Besides, We have answerd your final questions given that are very interesting to consider about this new topic

1) Line 34. Capital letter

        Done (Yellow colour in the text)

2)  Gender differences

        Done (Yellow colour in the text)

3) Mental health and overtraning should be discussed

        Done (Yellow colour in the text)

4) Fitness profile: age, gender,etc

        Done

5) Tools to quantify

        Done

Table 1. 

         This is a first approach in order to apply more personal programm for  team sports players

2.4 Evaluate these variables

         Individual somatotype, ADDED

2.5 Workouts, adjust, and how evaluate it?

         True, yes it is. We have added reference number 1. Goal standart test (Yellow colour in the text)

2.6 Previous sports experience?

Agree, thanks so mucc. DEFINITLY EXCELLENT IDEA. Probably the most important. WE HAVE ADDDED. (Yellow colour in the text)

2.7 Purpose of this pragraph

This part describes the modern approach about recovery process in order to accelerate based on biological responses. These paragraph is describeb acoording to author's own work  (REFERENCES 8, 37). Based on your question , we have added the referecence number 8. (Yelow colour in the text)

       How evaluate it?

Yes good think. We have included the reference number 8. In this article as first author, we described the frame work based on fatigue diagnosis, organization , protocolos and individual approach for recovery includes analysis about: responders or non responders, good or bad qualifiers (questionnaries) and intraindividual and interindividual variability).

       Questionnarie?

Yes for sure, among others

2.8, 2.9 Some topics about nutrition and health should be discussed?

        Done. (Yellow colour in the text)

Mental trianing ood or bad, how evalauted?

Good point, thanks so much, apreciatte it . We have included the criteria in the legend of table 1 part

(Yellow colour in the text) 

trap deaflite before?

We think that is a very good stimulus before that in order to prevent potential injuries

Table 2. strenght after training

We don't understand, in table number 2, there is no reference about it

Player 1. Power-resistance whole body?

Agree we have deleted. Done

Player 9. Too much on lower body?

Agree, we have changed

Done (Yelow colour)

Is posible in the real World?

This is a first apporach in order to be more individual inside the team sports system. In fact, durign the last years, all process are directed to personalize form.

The need personl supervision..?

In fact in the NBA and EUROLEAGUE, top players have their own team around them. This article could help to these team to be more efficient when they are working with top players. Besides, all authors are practicitioners and have experience in elite team sporst in basketball and soocer. 

Week or day?

Good point, the real World in team sports presents typicall weeks and  they are very importat. In fact there are weeks with 2,3 ,4 o 5 matches sometimes (for example NBA).

THANSK SO MUCH FOR YOUR COMMENTS

kING REGARDS

Reviewer 3 Report

The work is extremely interesting. Well illustrated.

Methodological techniques can be successfully used in the training and in the management of the training process.

Author Response

Dear Reviewer

Thanks so much for your kindly feedback. We really appreciate it

In advanced

King Regards

Round 2

Reviewer 2 Report

I am satisfied with the changes made. Only the part about the female triad should be changed (Line 138). It's not an injury, it's a syndrome.

Author Response

Dear Reviewe Number 3

Thanks so much for your respone and sorry for this minor detail. Probably we forgot it. Our intention was to consider all your apropiate comments

We have change it. (Green colour in the text), in this new version number 2

King Regards
